# Identification of Structural Variation in Chimpanzees Using Optical Mapping and Nanopore Sequencing

**DOI:** 10.3390/genes11030276

**Published:** 2020-03-04

**Authors:** Daniela C. Soto, Colin Shew, Mira Mastoras, Joshua M. Schmidt, Ruta Sahasrabudhe, Gulhan Kaya, Aida M. Andrés, Megan Y. Dennis

**Affiliations:** 1Genome Center, MIND Institute, and Department of Biochemistry & Molecular Medicine, Davis, CA 95616, USA; dcsoto@ucdavis.edu (D.C.S.); cshew@ucdavis.edu (C.S.); mnmastoras@ucdavis.edu (M.M.); gkaya@ucdavis.edu (G.K.); 2Integrative Genetics and Genomics Graduate Group, University of California, Davis, CA 95616, USA; 3UCL Genetics Institute, Department of Genetics, Evolution and Environment, University College London, London WC1E 6BT, UK; j.schmidt@ucl.ac.uk (J.M.S.); a.andres@ucl.ac.uk (A.M.A.); 4DNA Technologies Sequencing Core Facility, University of California, Davis, CA 95616, USA; rmsaha@ucdavis.edu

**Keywords:** structural variation, comparative genomics, chimpanzee, nanopore sequencing, optical mapping, chromatin organization, gene regulation, natural selection

## Abstract

Recent efforts to comprehensively characterize great ape genetic diversity using short-read sequencing and single-nucleotide variants have led to important discoveries related to selection within species, demographic history, and lineage-specific traits. Structural variants (SVs), including deletions and inversions, comprise a larger proportion of genetic differences between and within species, making them an important yet understudied source of trait divergence. Here, we used a combination of long-read and -range sequencing approaches to characterize the structural variant landscape of two additional *Pan troglodytes verus* individuals, one of whom carries 13% admixture from *Pan troglodytes troglodytes*. We performed optical mapping of both individuals followed by nanopore sequencing of one individual. Filtering for larger variants (>10 kbp) and combined with genotyping of SVs using short-read data from the Great Ape Genome Project, we identified 425 deletions and 59 inversions, of which 88 and 36, respectively, were novel. Compared with gene expression in humans, we found a significant enrichment of chimpanzee genes with differential expression in lymphoblastoid cell lines and induced pluripotent stem cells, both within deletions and near inversion breakpoints. We examined chromatin-conformation maps from human and chimpanzee using these same cell types and observed alterations in genomic interactions at SV breakpoints. Finally, we focused on 56 genes impacted by SVs in >90% of chimpanzees and absent in humans and gorillas, which may contribute to chimpanzee-specific features. Sequencing a greater set of individuals from diverse subspecies will be critical to establish the complete landscape of genetic variation in chimpanzees.

## 1. Introduction

Great apes have considerable phenotypic diversity despite being closely related species. For humans and chimpanzees, with only ~5 to 9 million years of independent evolution [1,2], significant effort has gone into understanding the underlying genetic and molecular differences contributing to species differences, often with the primary focus on human-unique features [3]. Direct comparison of protein-encoding genes has identified exciting candidates, but these only account for a minor proportion of species differences [4]. Recent analysis of Illumina short-read sequencing has allowed identification and genotyping of single-nucleotide variants (SNVs) at the genome scale, which have been used to address questions related to the demographic history and genetic adaptations of each species, and lineage-specific traits [5]. Further, transcriptome and epigenome comparisons of immortalized cell lines and tissues have revealed many thousands of individual genes and putative *cis*-acting regulatory elements that contribute to species differences in gene regulation [6,7,8,9,10,11,12,13], though often with varied results and reproducibility across studies. 

Since the publication of the chimpanzee genome [14], comparison with the human reference genome showed that structural variants (SVs), or genomic rearrangements such as inversions and copy-number variants (deletions and duplications), comprise a greater proportion of genetic differences than SNVs [15]. Though important, SVs are difficult to discover and genotype using traditional short-read Sanger and Illumina data. As such, genome-wide analyses of SVs have leveraged alternative approaches, including fosmid-end mapping [16], array comparative genomic hybridization (CGH) [17,18,19,20], digital array CGH using whole-genome shotgun sequencing of Sanger [21] and Illumina [22], and comparisons with improved genome assemblies [23,24,25,26]. Most recently, the advent of long-read sequencing technologies, capable of completely traversing variant breakpoints, has significantly facilitated discovery of novel SVs [27]. To date, only one study has performed long-read sequencing of a chimpanzee; the most recent improvement to the chimpanzee reference genome (panTro6) used hybrid long-read (PacBio) and long-range sequencing approaches (Bionanogenomics (BNG) and Hi-C) of one individual, Clint, a male representing the subspecies *Pan troglodytes verus*, significantly increasing the number of known SVs [26].

Recent comparisons of short- and long-read sequencing technologies using benchmark human genomic datasets revealed that multiple genomes [28] and combinatorial platforms [29] are required for comprehensive SV discovery; therefore, we performed long-range BNG optical mapping and Oxford Nanopore Technologies (ONT) long-read sequencing of additional chimpanzee individuals. These new datasets have allowed us to more comprehensively assess deletions and inversions in the chimpanzee genome. When compared with published whole-genome screens using orthogonal approaches, our approach validated existing variants and discovered many new variants. Knowing that SVs often alter gene functions and regulation [20,30], we characterized the association of our discovered SVs on differences in gene regulation and chromatin organization between human and chimpanzee, identifying a number of events that likely contribute to chimpanzee-specific differences.

## 2. Methods

### 2.1. Cell line Growth and DNA Extraction

Chimpanzee AG18359 and S003641 lymphoblastoid cell lines (LCLs) were generously shared with us by Dr. Yoav Gilad at the University of Chicago. LCLs were grown in T75 flasks with RPMI 1640 medium with L-Glutamine supplemented with 15% fetal bovine serum (Thermo Fisher Scientific, Waltham, MA, USA) and Penicillin-Streptomycin (100 U/ml, VWR, Radnor, PA, USA). For Illumina XTen sequencing, genomic DNA (gDNA) was isolated using DNeasy Blood and Tissue kit (Qiagen, Germantown, MD, USA) followed by RNase A treatment (Roche, Mannheim, Germany) and ethanol precipitation. For ONT PromethION sequencing, high molecular weight (HMW) gDNA was isolated from 5 × 10^7^ cells following a modified Sambrook and Russell method as described previously [26,31]. The integrity of the HMW DNA was verified on a Pippin Pulse gel electrophoresis system (Sage Sciences, Beverly, MA, USA). For the BNG assay, HMW gDNA was isolated from cells using the BNG Prep Blood and Cell Culture DNA Isolation Kit (BNG #80004). Briefly, 1.5 × 10^6^ cells were resuspended in Cell Buffer and embedded in an agarose plug. The plug was treated with Proteinase K for 18 h followed by RNase A digestion for one hour. After extensive washing, the plug was melted, agarose was digested, and drop dialysis was performed to clean the DNA. A Qubit dsDNA BR Assay kit (Thermo Fisher Scientific) was used to quantify the DNA. All sequence data generated as part of this project are available for download at the European Nucleotide Archive (accession number PRJEB36949). 

### 2.2. Determination of Chimpanzee Subspecies

gDNA isolated from AG18359 and S003641 LCLs was sequenced at ~30× coverage with Illumina HiSeq XTen (Novogene, Sacramento, CA, USA and the UC Davis Genome Center DNA and Expression Analysis Core, Davis, CA, USA, respectively) and SNVs were identified following a previously published approach [32]. Briefly, reads were mapped using BWA (v0.7.17) against the chimpanzee reference genome (CHIMP2.1.4) using BWA-MEM with default parameters. Picard (v2.18.23) MarkDuplicates was used to remove duplicates with the flag “REMOVE_DUPLICATES = true.” SNVs were called using FreeBayes (v1.2.0) with the following flags: “--standard-filters --no-population-priors -p 2 --report-genotype-likelihood-max --prob-contamination 0.05.” We then filtered autosomal SNVs with QUAL ≥ 30 and intersected with data from de Manuel et al. [32] callable genome regions, and finally merged with the 59 genomes from de Manuel et al. [32], using bcftools merge with the following flags: “--missing-to-ref --force-samples.” EIGENSOFT smartpca [33] was used to define principal components (PCs) using the 59 Great Ape Genome Project (GAGP) chimpanzee genomes [32] and the genomes from AG18359 and S003641 were projected onto these components. We estimated the variance explained by each of the first 20 PCs as the eigenvalue/sum (top 20 eigenvalues). To expedite the analysis, it was run on 50% of the genome-wide SNVs. Admixture analysis was performed with the software ADMIXTURE [34] with a set the number of ancestral populations *K* = 4 corresponding to the four chimpanzee subspecies. 

### 2.3. ONT Promethion Library Preparation and Sequencing

gDNA was sheared to an average size of 50 kbp using a Megaruptor instrument (Diagenode, Denville, NJ, USA) and then verified on a Pippin Pulse gel. A sequencing library was prepared starting with 2 µg of sheared DNA using the ligation sequencing kit SQK-LSK109 (ONT, Oxford, UK) following the instructions of the manufacturer with the exception of extended incubation times for DNA damage repair, end repair, ligation, and bead elutions. Thirty femtomole of the final library was loaded on PromethION R9.4.1 flow cell (ONT, Oxford, UK) and the data were collected for 64 h. Basecalling was performed live on the compute module using MinKNOW v2.1 (Oxford Nanopore Technologies, Oxford, UK). Details of the dataset can be found in Appendix A.

### 2.4. BNG Saphyr Library Preparation and Sequencing

AG18359 and S003641 were sequenced at the McDonnell Genome Institute at Washington University and the UC Davis Genome Center DNA and Expression Analysis Core, respectively. A total of 750 ng of HMW gDNA was labeled with DLE-1 enzyme, followed by proteinase digestion and a membrane clean-up step using the BNG Prep DLS DNA Labeling Kit (#80005). After overnight staining with an intercalating dye, the labeled DNA was loaded onto a Saphyr Chip G2.3 (BNG #20366) and run on the Saphyr system (BNG #60325) using the Saphyr Instrument Control Software (ICS, version 3.1) to maximize throughput of molecules. Raw images of DNA were converted into digital molecules files using Saphyr ICS version 3.1. Details of both datasets can be found in Appendix A. 

### 2.5. Detection of SVs

To detect SVs, ONT long-reads were mapped to the human (GRCh38, no alternative haplotypes) and the chimpanzee reference genome (panTro6) using minimap2 (v2.17-r941) and SVs were identified using Sniffles (v1.0.11) with “--genotype” flag and default parameters. Large SVs were identified from BNG opticals maps using Bionano Solve (v3.5) [35] *de novo* genome assembly and SV-discovery pipeline using human GRCh38 as the reference. The SV file in SMAP format was converted to VCF format using the smap_to_vcf_v2.py script contained in Solve software (v3.4.1). Only the variants with “PASS” filter were considered in the analysis and homozygous reference calls were removed. SV size selection and filtering were performed with the bcftools (v1.9) view using the filter “INFO/SVLEN ≥ 10,000 || INFO/SVLEN < −10,000” for both ONT and BNG datasets. To compare overlap between the SVs discovered by each method, we obtained 50% reciprocal overlap between features using bedtools intersect (v2.29.0) with flags “−f 0.5 −F 0.5.” Deletions and inversions were retrieved from the SVTYPE tag and processed separately in downstream analyses.

### 2.6. Genotyping and Filtering of SVs

Variants for each callset were genotyped independently using previously published Illumina data from 25 chimpanzees from all four subspecies, as well as eight gorillas and eight humans. SNV genotypes from non-human primates were retrieved from the GAGP [5] and human SNV genotypes were obtained from the Simons Genome Diversity Project [36] (Appendix A). Reads were mapped to the human reference (GRCh38) using BWA MEM (0.7.17−r1188) [37] and subsequently merged and sorted with samtools (v1.9) for each individual. Large inversions and deletions (>10 kbp) were genotyped with SVtyper (v.0.7.1) [38]. Genotype information was retrieved using bedtools query (v2.29.0). To assess whether a variant was novel to this study, calls were compared to previously reported deletions and inversions larger than 10 kbp found in any great ape or any variant discovered in chimpanzee [22,23,26] using bedtools intersect (v2.29.0) with 50% reciprocal overlap. SVs that were either (1) genotyped in one chimpanzee individual (1/1 or 0/1) or (2) reported as discovered in chimpanzee in previous studies, were selected to generate a higher confidence set (filter 1). This dataset was further refined by collapsing calls within the dataset with 50% reciprocal overlap. All novel calls were visually inspected in Integrative Genome Browser for ONT calls [39] and Bionano Access for BNG calls. Also, SVs present in ≥90% of the chimpanzee individuals (22 or more) as well as absent in outgroups (human and gorilla) were included in the likely chimpanzee-specific dataset (filter 2). In Kronenberg et al. [26], eight chimpanzee individuals were genotyped; as such, variants with evidence in seven or more individuals were also included in the chimpanzee-specific dataset. The distribution of high-confidence calls across the human reference (GRCh38) was plotted using the R package Karyoplotter [40].

### 2.7. Annotation of Impacted Genes

Genes impacted by SVs were obtained by intersecting Gencode v27 genomics features annotation file to deletion coordinates ±2.5 kbp and inversion breakpoints (considered as estimated breakpoints ±2.5 kbp and ±50 kbp) using bedtools intersect (v2.29.0). The impact of the SVs on the function of the gene was predicted using Ensembl Variant Effect Predictor (VEP) [41] with the Gencode v27 GTF file. The probability of loss of function intolerance score (pLI) was obtained from the gene constraints scores table in the Exome Aggregation Consortium database [42]. Gene ontology (GO) annotations and overrepresented terms were retrieved for each gene using DAVID [43,44] and by selecting terms at a 5% false-discovery rate (FDR). Genes previously identified as showing signatures of positive and balancing selection in chimpanzees were retrieved from previously published data [45], and intersected with the set of genes impacted by SVs.

### 2.8. Differential Gene Expression

We obtained previously-published RNA-seq data from chimpanzee and human LCLs [7] and induced pluripotent stem cells (iPSCs) [46]. Raw data were trimmed using TrimGalore (v0.6.0) with the following parameters: “-q 20 --phred33 --length 20”. Transcripts per million (TPM) values were estimated using Salmon (v0.14.1) [47] with the “--validateMappings” flag for all transcripts in GENCODE v27 and chimpanzee transcriptome published by Kronenberg et al. (2018) [26], which was based on a combination of orthologous genes identified via comparisons of human GENCODE v27 and novel transcripts identified through PacBio isoSeq of iPSCs. The R package tximport [48] was used to estimate gene-level counts from TPM values using the setting ‘countsFromAbundance = “lengthScaledTPM”’ for 55,461 annotated genes with equivalent identifiers in the two transcriptomes. Differential expression analysis was conducted with limma-voom [49,50]. Genes with fewer than 1 count per million across all samples were filtered from the analysis, and a model accounting for species and sex was implemented. Differentially-expressed (DE) genes were called at a 5% FDR.

### 2.9. Topologically-Associated Domain (TAD) Analyses

We retrieved published TAD predictions from an LCL of a human female (GM12878) originally called with 4.9 billion Illumina reads [51]. Domain coordinates were transformed from GRCh37 to GRCh38 using liftOver (UCSC Genome Browser; 9262/9274 domains successfully converted). Boundaries were defined as the start and end coordinates of each domain expanded to 5 kbp (resolution size of the TAD-calling analysis).

To directly compare domain boundaries between humans and chimpanzees, we generated DNase Hi-C libraries from three human (GM12878, GM20818, GM20543) and two chimpanzee (S007602, AG18359) LCLs as described by Ramani et al. [52]. Raw data were processed using the Juicer pipeline [53] with the human reference GRCh38. Human alignments were downsampled to ~300 million reads to allow for equal comparison to chimpanzee, and Hi-C interaction matrices were generated with a (BWA) MAPQ filter of 30. Domains were called on Knight-Ruiz normalized contact matrices using TopDom [54] at 50 kbp resolution and the default window size (w = 5). Similarity between domain sets was computed with the Measure of Concordance (MoC) as implemented previously [55] using chromosome 1. Domain calls were visualized with interaction maps (coverage normalized at 5 kbp resolution) using Juicebox (1.11.08). Across all chromosomes, boundaries unique to each species were considered to be the left and right coordinates of each domain, expanded to 50 kbp, when that region was not adjacent to (or overlapping) a boundary from the other species. This analysis was repeated using high-depth raw Hi-C data from four human and four chimpanzee iPSCs with approximately 1 billion reads per sample (combined across individuals; also normalized by downsampling) [12].

### 2.10. Permutation Analyses

For each variant, the distance to the nearest segmental duplication (SD; duplicated region with >90% identity across >1 kbp, downloaded from UCSC Genome Browser GRCh38) was calculated using bedtools closest (v2.29.0). Regions of the same size (deletions ±2.5 kbp and inversions ±2.5 kbp) were randomly sampled from the human genome using bedtools shuffle (v2.29.0), and 5-kbp “breakpoints” were extracted from shuffled inversions. The distribution of the distance of these random regions to the nearest SD was plotted as density using the R package ggplot2. Permutation tests to assess the enrichment/depletion of genomic features (e.g., genes, boundaries) at SVs were similarly performed by shuffling the SV coordinates 1000 times and counting the number of intersecting features with each set of coordinates. SVs were tested for enrichment of DE genes by generating 1000 random samples of all genes tested in the expression analysis of equal size to the differential set. One-tailed empirical *p*-values were calculated as follows: *p*-value = (M + 1)/(N + 1), where M is the number of iterations yielding a number of features less than (depletion) or greater than (enriched) observed and N is the number of iterations.

## 3. Results

### 3.1. Large-Scale SV Discovery and Genotyping in Chimpanzee

To date, one western chimpanzee individual (Clint) comprising the reference genome (panTro6) has been subject to hybrid long-read sequencing for genome assembly and SV discovery [26]. We sought to expand SV discovery via long-read sequencing to two additional chimpanzee individuals (AG18359 and S003641) for which renewable LCLs and functional genomic information, including RNA-Seq and ChIP-Seq data [7,13,56], are available. To begin, we performed Illumina short-read sequencing (~30× coverage) of both individuals to confirm ancestry via SNV detection followed by comparisons of population-specific genetic markers and PC analysis with chimpanzees from the GAGP [5] (Appendix A). From this, we determined AG18359 to be a female western chimpanzee (*Pan troglodytes verus*) and S003641 to be a male western chimpanzee with some central chimpanzee ancestry (*Pan troglodytes verus* × *Pan troglodytes troglodytes*). Notably, ~13% of the ancestry of this individual is assigned to the central-chimpanzee population, similar to one individual (Donald) that was sequenced as part of the GAGP.

To discover potentially novel chimpanzee SVs, we assayed AG18359 gDNA using ONT PromethION (29×) and BNG optical mapping (116×) (Appendix A). To compare SV discovery of two individuals on the same platform, we also subjected S003641 to BNG optical mapping (70×). As it is the most accurate and well-annotated primate assembly, we mapped our sequence data to the human reference genome (GRCh38). We excluded SDs and insertions from our analysis of SVs due to challenges in their discovery and validation [57]. Focusing exclusively on deletions and inversions, we discovered 49,579 deletions and 560 inversions using ONT and 4790 deletions and 280 inversions using BNG from AG18359. Similarly, we identified 5407 deletions and 207 inversions using BNG from S003641. For comparative purposes, we also mapped the AG18359 ONT sequence data to the most recent chimpanzee reference genome (panTro6) and discovered fewer events (7895 deletions and 142 inversions) suggesting that a significant proportion of SVs identified via mapping to the human reference represented species differences. 

As the primary goal of our study was to identify species differences, we moved forward with SVs identified using the human reference genome. We next compared SV discovery across our two platforms. Although ONT had higher sensitivity to discover smaller variants, down to 50 bp, there was a higher chance of detecting false positives and errors at this resolution (Appendix A). To properly compare across technologies, we filtered for large SVs (≥10 kbp) and compared similarities by consolidating variants with more than 50% reciprocal overlap. We found a comparable number of deletions in our three call set (586, 586, and 666 events in AG18359 ONT, AG18359 BNG, and S003641 BNG, respectively) with 138 deletions found by all three call sets (Appendix A, Appendix A). Out of the 586 deletions found in the AG18359 ONT call set, 381 were uniquely discovered using this technology, while BNG contributed another 553 deletions, out of which 307 (55.5%) had support from both individuals. As such, deletion call sets from the same technology exhibited a greater overlap than comparing calls from different technologies of the same individual. We also found a comparable number of inversions across all three call sets (243, 269, and 207 variants in AG18359 ONT, AG18359 BNG, and S003641 BNG, respectively) (Appendix A), of which 34 variants were shared among them all. Again, the most overlap for inversions was identified between different individuals assayed using the same BNG technology, representing 80 shared out of the total 274 unique variants.

In order to narrow in on a higher-confidence set of SVs, we subsequently performed genotyping of this discovery set using short-read Illumina data from GAGP (>20-fold coverage) of all four chimpanzee subspecies (*n* = 25) (Appendix A) using SVTyper [38]. We also compared our discovered SVs with previously-reported datasets from three recent whole-genome SV screens of chimpanzees [22,23,26], each using diverse genomic methods for discovery (Appendix A). From this, we identified 425 deletions and 59 inversions that had support from short-read genotyping and/or intersecting with a previously-discovered SV (Appendix A). In all, our discovery approach using ONT and BNG data achieved 88 novel deletions and 36 novel inversions when compared with the most recent genome-assembly alignment [23,26] and read-depth [22] approaches (Figure 1A,B). 

### 3.2. Genomic Features of Identified SVs

Examining genomic features of our high-confidence set of chimpanzee SVs, we found that deletion sizes ranged between 10 kbp (our minimum threshold) up to ~526 kbp (31 kbp mean; 18.5 kbp median) (Figure 1C) and inversions ranged in size between 10 kbp and 78 Mbp (4.1 Mbp mean; 57.3 kbp median), including four of seven known chimpanzee pericentric inversions identified only with ONT (*n* = 2) or with both technologies (*n* = 2) [58,59,60,61,62,63,64]. The majority of novel inversions identified in our study tended to be smaller (57 kbp mean length), perhaps influenced by strict size cutoffs (>100 kbp) used in previous studies [23]. The distribution of SVs across the human genome (Figure 1A and Appendix A) was relatively uniform for deletions, which were found on all 24 chromosomes. The greatest number of events were identified in chromosome 2 (*n* = 34); however, when normalizing by the total number of bases, chromosomes 19 (0.34 deletions per Mbp) and 21 (0.32 deletions per Mbp) exhibited the highest number of deletions (Appendix A). Inversions, on the other hand, were found on 19 chromosomes, with chromosome 5 exhibiting the greatest number of variants (*n* = 8), and chromosomes 5, 7 and 12 displaying the greatest number of inversions per chromosome size (0.04 inversions per Mb). Further, we found that SV breakpoints of both deletions and inversions were non-randomly distributed across the human genome near SDs (Figure 1D, empirical *p*-value = 1 × 10^−4^), similar to previously reported results for distribution of SDs in primate genomes [21,22,65,66]. This observed clustering may be accounted for by SD-mediated deletions and inversions that can be created via non-allelic homologous recombination [67]. 

### 3.3. Genes Impacted by SVs

To evaluate the functional impact of our high-confidence set of SVs, we retrieved all annotated transcribed features within deletions (±2.5 kbp) and at inversion breakpoints (±50 kbp) (Appendix A). Deletions overlapped with 592 genes, out of which 162 were protein-encoding genes (Figure 2A). To further refine the impact of SVs and gene function, we focused on protein-encoding genes and used Ensembl Variant Effect Predictor (VEP) to predict functional impact. VEP annotated 80 protein-encoding genes as highly impacted by deletions (i.e., feature ablation or truncation), out of which 54 have been previously classified as loss of function (LoF) tolerant (pLI ≤ 0.1) by the Exome Aggregation Consortium [68,69] (Figure 2B). Also, three genes (*ATXN2L*, *SH2B1*, and *IL27*), which all reside within the same ~500 kbp “deletion” mapped to human chromosome 16p11.2, were classified as LoF intolerant (pLI ≥ 0.9). A search through the chimpanzee reference (panTro6) found *ATXN21* and *SH2B1* residing on an uncharacterized chimpanzee chromosome Un_NW_019937196v1, suggesting that these genes have been translocated to a new genomic locus. This is likely the case for other genes with predicted high-variant effect and LoF intolerance. Focusing on inversions, we found breakpoints overlapping with 342 transcribed elements of which 64 genes were within 2.5 kbp of breakpoints, including 95 and 21 protein-encoding genes, respectively (Figure 2A). No highly impacted genes, as predicted by VEP, were found in this dataset. Using pLI scores, we identified 9 genes either modified or overlapped by inversions classified as loss-of-function intolerant in humans (Figure 2B). 

In total, we found a significant depletion of protein-encoding genes at deletion regions (162 genes within 2.5 kbp, empirical *p*-value = 0.001, Figure 3 and Appendix A) as well as at inversion breakpoints (21 protein-encoding genes within 2.5 kbp, empirical *p*-value = 0.001, Figure 3 and Appendix A). Notably, this depletion did not persist when considering all transcribed elements intersecting SVs. Taking a closer look at genes with clear orthologs between chimpanzee and humans, we identified significantly fewer orthologs of deletion-impacted genes vs. inversion-impacted genes (67% vs. 89%, respectively; *p*-value = 1 × 10^−5^ Fisher’s exact test). The majority of deletion-impacted genes with no orthologs were predicted to have high-VEP effect (179 out of 195 genes), suggesting that deletion of these genes completely ablated them from the chimpanzee genome. 

Finally, we explored functional annotations of genes impacted by SVs. We found 208 transcribed elements impacted by deletions with known GO annotations as reported by DAVID [43,44] (Figure 2C). Compared to the complete set of human GO annotations, this gene list displays an overrepresentation of genes associated with sensory perception of smell (GO: 0050911, *q*-value = 8.7 × 10^−11^ and GO:0007608, *q*-value = 3.3 × 10^−2^). We also found an overrepresentation of deletion-impacted genes involved in the G-protein coupled receptor signaling pathway (GO: 0007186, *q*-value = 5 × 10^−5^). Notably, both ontologies are primarily driven by known copy-number polymorphism that exists among olfactory-receptor genes [70]. Inversions contained 140 genes with known GO functional annotation exhibiting an overrepresentation of regulation of cell differentiation (GO: 0045596, *q*-value = 1.2 × 10^−4^).

### 3.4. SVs and Gene Regulation

To understand if variants might affect gene regulation, we leveraged existing RNA-seq datasets generated from chimpanzee and human LCLs [7] and iPSCs [46]. From 55,461 human–chimpanzee orthologous transcribed features, we identified 6565 and 8946 genes in LCLs and iPSCs, respectively, as significantly DE between the two species (Appendix A). Among genes for which human-chimpanzee orthology was assigned that directly intersected SVs (*N* = 397 in deletions ±2.5 kb; *N* = 61 for inversion breakpoints ±2.5 kb), roughly half were significantly DE (57/135 LCL and 60/129 iPSC tested genes in deletions; 25/37 LCL and 22/36 iPSC tested genes in inversion breakpoints) (Appendix A). We report a significant enrichment of DE genes from both cell types within (±2.5 kb; permutation test empirical *p*-value < 0.04) and near (±50 kb; *p*-value < 0.01) deletions and near (±50 kbp; *p*-value < 0.002) inversion breakpoints. DE gene enrichment was only significant within (±2.5 kbp) inversion breakpoints in LCLs Figure 3 and Appendix A).

Considering that gene regulation may be affected by changes in genome organization, we next assayed the impact of SVs on chromatin structure by intersecting with previously identified TADs from a deeply-sequenced human LCL (GM12878) [51] and found 45 and 17 TAD boundaries likely disrupted by deletions and inversions, respectively, in chimpanzees. Similar to what others have reported [71,72], deletions were less likely than expected by chance to straddle TAD boundaries, thereby generating putatively disrupted TADs (PDTs) (permutation test empirical *p*-value < 0.01 within 2.5 kbp and 50 kbp of deletions; Figure 3 and Appendix A). This is consistent with the hypothesis that regions maintaining chromatin structure are subject to negative selection. Not previously reported, we also found a significant depletion of PDTs intersecting inversions (*p*-value = 0.001 within 2.5 kbp and 50 kbp of inversions; Appendix A). Within PDTs we identified 58 and 65 DE genes in LCLs and iPSCs, respectively. This suggests that disruption of genome organization may have contributed to interspecies changes in gene expression for a subset of genes. Example loci are highlighted in Figure 4A, Appendix A. Notably, chromatin structure was also apparently altered by variants near but not directly intersecting identified TAD boundaries (Figure 4B and Appendix A).

To examine chromatin structure of PDTs, we generated orthologous Hi-C maps from human and chimpanzee LCLs and iPSCs [12] against the human reference (GRCh38) and directly compared differences in domain boundaries between species. Overall, domain calls were similar between species (MoC 0.75 and 0.79 for LCLs and iPSCs, respectively [55]). We examined chimpanzee PDTs and identified more chimpanzee-unique boundaries than genome-wide boundaries (30.5% (18/59) versus 24.9% (1424/5714)). Similarly, for iPSCs we found 22.0% (13/59) of boundaries in PDTs were not shared with human, compared to 14.9% genome-wide boundaries (868/5834). These numbers suggest that TAD-altering SVs may impact chromatin structure in chimpanzees. 

Closer inspection of these regions revealed examples of altered gene expression coinciding with changes to three-dimensional chromatin structure. For example, the breakpoints of an inversion mapping to human chromosome 2q12.2-13 lie near altered domain boundaries and DE genes in iPSCs. Both *UXS1* and *SH3RF3* reside in altered domains and show increased contact frequency with chimpanzee-proximal inverted sequences that are over 1 Mbp away in the human genome (Figure 4A and Appendix A). Similar gains of interactions are visible in the LCL Hi-C data with *UXS1* also DE, though in the opposite direction (Appendix A). A smaller inversion mapping to human chromosome 9q22.31 appears to mediate a domain fusion in both iPSCs and LCLs (Figure 4B and Appendix A). In both cell types, the nearby (<8 kbp away) gene *SPTLC1* and truncated processed pseudogene *AL136097.2* are upregulated and downregulated, respectively, in chimpanzees compared with humans (Figure 4B and Appendix A). Other examples of domain-altering deletions and nearby DE genes are presented in Appendix A. Altogether, these data provide evidence that SVs may drive DE patterns, either through disruption of the transcribed sequence itself or through altered *cis*-acting regulation, mediated by reorganization of physical interactions within chromatin.

### 3.5. Genes Showing Signatures of Natural Selection

Recent efforts to sequence diverse great ape genomes have led to identification of signatures of natural selection using SNV data that may help to explain features unique to chimpanzee species and subspecies [5,32,45,73]. To understand if our identified SVs might impact the outcome of such studies or explain signatures of selection previously identified, we compared our map of SVs with a recent study of natural selection in multiple genomes of the four chimpanzee subspecies (*Pan troglodytes verus*, *troglodytes*, *ellioti*, and *schweinfurthii*) mapped to the human reference genome [45]. In this study, among several other tests, the Hudson–Kreitman–Aguade (HKA) test [74] was used to identify the top 200 genes showing the strongest signatures of long-term balancing selection and positive selection in each subspecies. Intersecting this set of genes with our complete list of genes residing within or near deletions (Appendix A), we determined that of the 592 genes putatively disrupted by a deletion, 54 show strong signatures of natural selection using the HKA test (32 for positive and 22 for balancing selection). For inversions, of the 342 genes at or near inversion breakpoints, six show strong signatures of natural selection (five for positive, one for balancing) (Appendix A). Of all the genes affected by SVs and with strong signatures of natural selection, nine have evidence of DE in either LCLs or iPSCs, including two protein-encoding genes showing signatures of balancing selection: *INPP4B*, which carries a deletion upstream of the transcription-start site and is upregulated in chimpanzee LCLs, and *HLA-F*, which is completely deleted and is upregulated in chimpanzee LCLs and downregulated in iPSCs. The possibility that these deletions generated beneficial expression changes that became strongly affected by natural selection makes these genes interesting candidates for follow up. 

### 3.6. Genes Impacted by Chimpanzee-Specific SVs

To hone in on SVs unique and universal to chimpanzees that may contribute to species-specific features, we consolidated the complete dataset of our newly discovered SVs and those previously published [22,23,26]. Filtering for only those with positive genotypes in >90% of chimpanzee individuals genotyped but found in neither humans (*n* = 8) nor gorillas (*n* = 8), we identified 209 deletions and 18 inversions. This set ranged in size from 10 kbp to 526 kbp for deletions and 12 kbp to 78 Mbp for inversions (including the four large-scale cytogenetic events). Again due to the olfactory receptors at these loci, GO analysis shows that the genes contained within these SVs were overrepresented for the detection of chemical stimulus involved in sensory perception of smell (GO:0050911, *q*-value 4.1 × 10^−2^). Focusing on genes with a higher likelihood of being functionally impacted by SVs, we identified 56 protein-encoding genes with a high-impact VEP score (deletions) or within 2.5 kbp of a breakpoint (inversions) (Table 1). Of the 35 genes queried in our cross-species RNA-seq comparisons, 13 exhibited significant DE in chimpanzee versus human in LCLs and/or iPSCs, including *APOL4*, *CAST*, *CLN3*, *EFCAB13*, *EIF3C*, *IL18R1*, *NPIPB8*, *NPIPB9*, *NUPR1*, *RABEP2*, *SGF29*, *SLC01B3*, and *SULT1A1*. Additionally, six genes showed strong signatures of positive selection (*APOBR*, *IL27*, and *TUFM* at human chromosome 16p11.2 and *OR10H1* and *OR10H5* at human chromosome 19p13.12) or balancing selection (*CLC* at human chromosome 19q13.2). In all, this list of genes represents exciting candidates putatively implicated in chimpanzee-specific traits. 

## 4. Discussion

Most extensive SV analyses using comparative genomic approaches have used a single genome from one chimpanzee individual of the subspecies *Pan troglodytes verus* (i.e., Clint) [14,16,21,23,24,26]. Here, we performed long-read sequencing of two additional individuals of the same subspecies, one of which carried admixture with *Pan troglodytes troglodytes*, using two orthogonal technologies: optical mapping and nanopore sequencing. To our knowledge, this represents the first nanopore sequence of a chimpanzee genome. From this, we discovered over 60,000 deletions and over 500 inversions (≥50 bp) when compared with the human reference (GRCh38), on the same scale as found in a recent comparison of the new chimpanzee assembly using a hybrid assembly approach (panTro6) [26]. As expected, ONT sequencing was capable of detecting significantly more SVs, down to 50 bp with higher resolution at breakpoints (Appendix A), compared to our BNG datasets. Many of the bioinformatically-identified SVs were redundant within and across technologies, which required additional filtering. To determine a higher-confidence set of SVs, we limited our analysis to variants ≥10 kbp in size with short-read Illumina sequencing evidence of the variant using SVtyper, a genotyping approach. Though the genotyping step significantly increased our confidence in variant calls, it also reduced the number of variants we identified (from 1838 to 858 deletions and from 719 to 253 inversions), particularly for inversions, which are difficult to detect/genotype using short-read data. Additionally, our strict size cutoff limited our ability to discover transposable elements, which has been shown to represent a significant proportion of lineage divergence between chimpanzees and humans [75]. Furthermore, due to the uncertainty of the BNG breakpoints, most SVs discovered using only this approach were largely filtered from our subsequent analyses due to an inability to accurately genotype events. Nevertheless, our approach led to the discovery of 88 novel deletions and 36 novel inversions when compared to recent genome-wide scans. We note that we also excluded SDs and insertions from our analysis due to difficulties in discovery and subsequent validations using standard short-read genotyping approaches [76]. As improved hybrid-based methods combining long- and short-read data are developed to more accurately identify SVs and their breakpoints, it will be a worthwhile endeavor to return to our dataset to discover additional SVs.

Our results implicated chimpanzee SVs in potentially impacting gene regulation and chromatin organization. It has been established that TAD structures are evolutionarily conserved [51,77], and recent work finds that deletions altering TAD boundaries in humans are under purifying selection [71,72]. TAD structure is also conserved across apes, as evidenced by the incidence of gibbon–human synteny breaks at domain boundaries [78]. Similarly, we find a depletion of PDTs generated by deletions in chimpanzees, as well as an expected but previously unreported reduction of inversions altering TADs. Taken together, the paucity of SVs altering domain boundaries suggests such variants in chimpanzee experience strong negative selection, as observed in other species, perhaps due to conserved roles of TADs in modulating gene regulation. Despite the overall depletion of SVs at TAD boundaries, we did find an increased incidence of species-specific domain boundaries and significant enrichment of DE genes near SVs in the two cell types queried in this study, concordant with previous findings assessing the impact of deletions and duplications on differential gene expression in primate LCLs [20]. Our analyses are subject to some limitations. Domain calling is highly sensitive to input parameters, but the pairs of Hi-C maps were subject to the same analysis and highly correlated at a variety of resolutions tested (MoC > 0.7 at 100 kbp, 50 kbp, 25 kbp, and 10 kbp for iPSCs; 100 kbp and 50 kbp for LCLs) allowing for an assessment of genome-wide domain differences. Though the number of aligned reads were normalized to comparable levels, relative read depth is likely to vary across the genome due to differences in mappability. This is particularly likely at SV loci, where deletions and SDs generate discontinuities in the Hi-C matrix. As such, these domain calls should be interpreted primarily as a means of identifying regions of putatively disrupted chromatin structure. 

Notably, many of the genes near SVs were not DE; however, it is plausible that these non-DE genes either remain connected to their regulatory elements or their associated elements are specific to cell types not assayed. Further, while it has been reported that topology-altering SVs can have little effect on gene expression [79], or that expression is not globally altered by loss of TADs [80], it could still be the case that expression-altering SVs are frequently subject to negative selection. For instance, TAD- and expression-altering SVs reported in humans are typically *de novo* and pathogenic [81,82]. Regardless, our findings are concordant with those of Kronenberg et al. (2018) [26], who reported an enrichment of human–chimpanzee cortical organoid DE genes near fixed human-specific SVs. While they find an enrichment for downregulated genes at insertions and deletions and upregulated genes at SDs, their analysis produced a much smaller set of DE genes (785 across both cell types using single-cell RNA-seq) and a much larger set of variants (17,789). These findings are also in line with reports that SVs underlie many human expression quantitative trait loci [83]. However, considering the currently incomplete understanding of the relationship between gene regulation and three-dimensional chromatin structure, we emphasize that functional studies are necessary to causally implicate SVs in gene expression differences within or between species.

In addition to using Illumina genotyping of our identified SVs to filter out putatively false positive variants, we also used this information to query SV differences across subspecies. In our high-confidence set of SVs, we identified one novel deletion in chimpanzees (human chromosome 6q11.1; chr6:60639753-60662981, GRCh38) from our BNG data of the western individual carrying substantial central ancestry (S003641) that was also found uniquely in central chimpanzees (*n* = 4). Considering the relatively low ancestry contribution of this individual assigned to the central-chimpanzee population (~13%), this highlights the importance of sequencing more diverse individuals to identify additional subspecies-specific SVs to better survey the complete variant landscape. Using these same genotypes, we also focused on a set of genes universally impacted by SVs across all chimpanzees tested, but not detected in the other great apes studied (humans and gorillas), since these genes may putatively contribute to species-specific traits (Table 1). One example, *APOL4*, encoding Apolipoprotein L4, was completely deleted in all chimpanzees tested (*n* = 25) and also shown to be downregulated in both LCLs and iPSCs in chimpanzees when compared with humans. This gene is a member of a tandemly-duplicated family that has experienced a recent expansion in the primate lineage [84] and may play a role in lipid trafficking throughout the body. Human polymorphism at this locus has been shown to be associated with schizophrenia [85]. Several identified genes also exhibited signatures of natural selection. One example region putatively under balancing selection includes two deletions impacting the primate-expanded galectin gene cluster, a family of proteins that specifically bind β-galactoside sugars and are important in modulating immune response through interactions with T cells [86]. Both deletions (10 kbp and 35 kbp in size, respectively) are found homozygously in all chimpanzees tested (*n* = 25), and thus are likely not the target of balancing selection, but they completely ablated *CLC* (or *LGALS10*) and *LGALS14*, as well as the downstream region of *LGALS13* (Appendix A). Two of these genes (*LGALS13* and *14*), expressed exclusively in human placenta [87], are important drivers of maternal adaptive immune response, with reductions in expression of either gene shown to be associated with an increased risk of preeclampsia [88]. Although the mechanisms are unclear, it is notable that other immune-related genes with connections to preeclampsia also exhibit signatures of balancing selection in humans [89,90,91]. It is possible that deletions impacting this gene cluster may contribute to pregnancy-related outcomes in chimpanzees that could be subject to natural selective pressures. 

## Figures and Tables

**Figure 1 genes-11-00276-f001:**
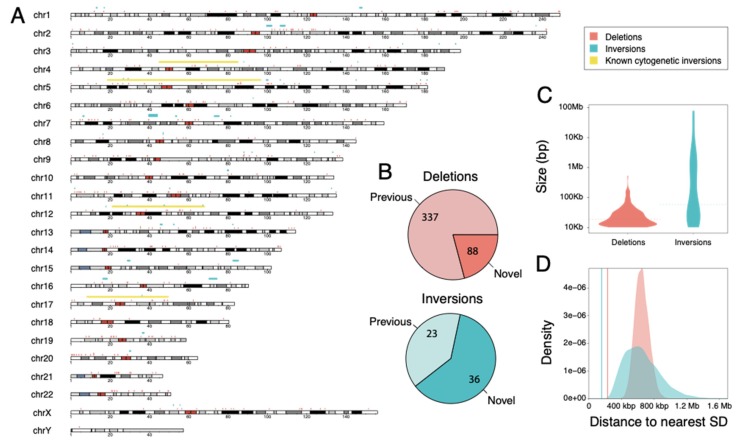
Genomic features of identified SVs. (**A**) Deletions (red), inversions (cyan), and large-scale cytogenetic inversions (yellow) are interspersed across all 24 human orthologous chromosomes, depicted as ideograms. (**B**) Novel variants in our dataset defined as lacking 50% reciprocal overlap with previous reported variants in great apes. (**C**) Size distribution of deletions (red) and inversions (cyan). Median size is depicted as dashed lines. (**D**) Observed average distance of deletions (red line) and inversions (cyan line) to SDs, compared to randomly sampled regions across the genome of the same size of deletions (red distribution) and inversion (green distribution). We observed an enrichment of SV breakpoints residing near SDs (empirical *p*-value = 1 × 10^−4^).

**Figure 2 genes-11-00276-f002:**
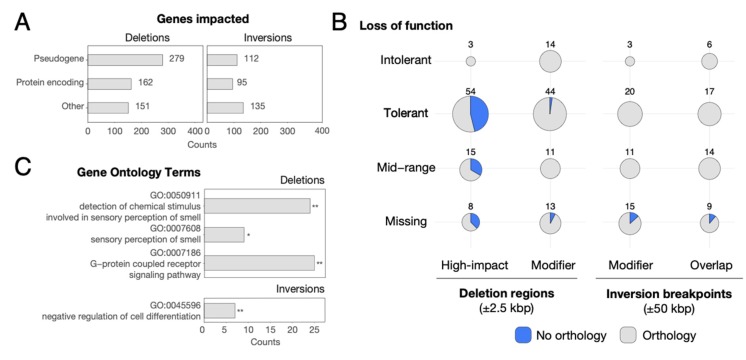
Description of genes overlapping identified SVs. (**A**) Categories of genes overlapping deletion regions ±2.5 kbp and inversion breakpoints ±50 kbp as defined by ENSEMBL biotypes. (**B**) Number of protein-encoding genes classified as LoF tolerant (pLI ≤ 0.1), intolerant (pLI ≥ 0.9) and middle range (pLI > 0.1 and pLI < 0.9) affected by deletions regions ±2.5 kbp and inversion breakpoints ±50 kbp. Some affected genes lack LoF information (missing category). All genes impacted by deletions were classified by VEP as either highly impacted (feature ablation or truncation) or modified, while genes impacted by inversions were either modified or no effect was predicted (overlap only). Transcribed elements with no corresponding ENSEMBL transcript ID in humans were classified as no orthology (blue). (**C**) Overrepresented GO terms in genes impacted by deletions and inversions as reported by DAVID (* *q*-value < 0.05; ** *q*-value < 0.001). Counts represent the number of genes annotated with each GO term.

**Figure 3 genes-11-00276-f003:**
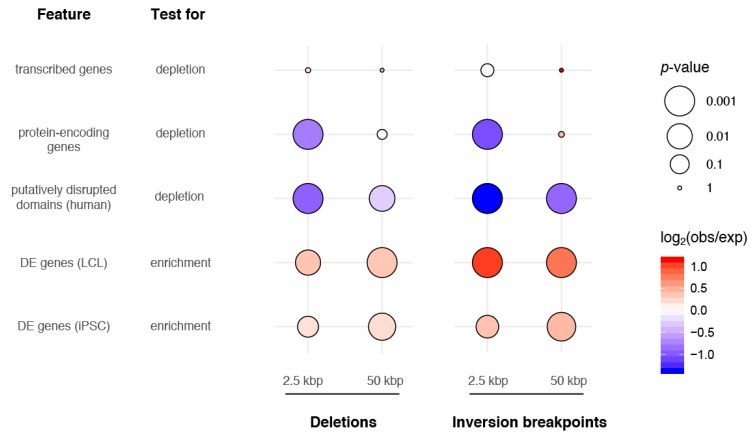
Enrichment and depletion tests of SVs with genomic features. Both deletions and duplications were tested within 2.5 kbp (resolution of the SV calls) and 50 kbp. All annotated genes (GENCODE v27) and protein-encoding genes were tested for depletion of SVs (top two rows) via permutation testing. Human TADs from the LCL GM12878 were tested for depletion of putatively disrupting SVs (i.e., SVs generating PDTs, third row). Human–chimpanzee DE genes from LCLs and iPSCs were also tested for enrichment in SVs via permutation testing (fourth and fifth rows). Circles are sized proportionally to the negative log of the empirical *p*-values and colored according to the strength of enrichment or depletion, represented by the log ratio of observed (obs; number of features intersecting SVs) and expected (exp; mean number of features intersecting 1000 permuted coordinate sets) counts.

**Figure 4 genes-11-00276-f004:**
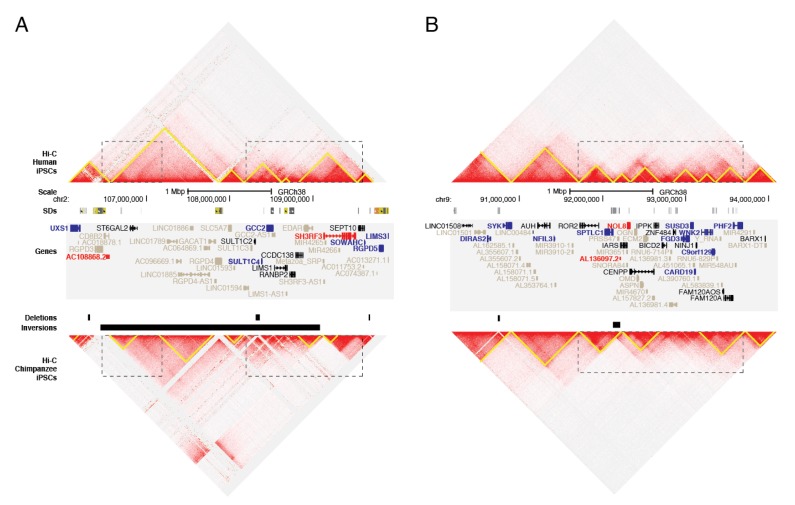
Genome organization of human and chimpanzee across regions with identified SVs. The Hi-C genomic landscape of human (top) and chimpanzee (bottom) are depicted for iPSCs using Juicebox for (**A**) chromosome 2q12.2-q13 (chr2:106095001-109905000, GRCh38) and (**B**) chromosome 9q22.2-q22.32 (chr9:90200001-94010000, GRCh38). Predicted TADs (yellow triangles) were compared between species, noting differences at SVs (dotted boxes) including deletions and inversions. SDs are depicted as colored bars, taken from the UCSC Genome Browser track. Genes showing significant DE in chimpanzee versus humans are colored as red (up in chimpanzee) or blue (down in chimpanzee). Genes not included in the DE analysis are in gray (Appendix A).

**Table 1 genes-11-00276-t001:** Protein-encoding genes impacted by chimpanzee-specific deletions and inversions.

Gene	ENSEMBL ID	SV Type	Description
***APOBR***	**ENSG00000184730**	**deletion**	**Apolipoprotein B receptor**
*APOL1*	ENSG00000100342	deletion	Apolipoprotein L1
*APOL4 **	ENSG00000100336	deletion	Apolipoprotein L4
*ATP2A1*	ENSG00000196296	deletion	Sarcoplasmic/endoplasmic reticulum calcium ATPase 1
*ATXN2L*	ENSG00000168488	deletion	Ataxin 2 like
*CARD18*	ENSG00000255501	deletion	Caspase recruitment domain family member 18
*CAST **	ENSG00000153113	inversion	Calpastatin
*CD19*	ENSG00000177455	deletion	CD19 Molecule
*CEACAM21*	ENSG00000007129	deletion	CEA Cell Adhesion Molecule 21
*CFHR2*	ENSG00000080910	deletion	Complement Factor H Related 2
*CFHR4*	ENSG00000134365	deletion	Complement Factor H Related 4
***CLC***	**ENSG00000105205**	**deletion**	**Charcot-Leyden crystal Galectin**
*CLN3 **	ENSG00000188603	deletion	CLN3 Lysosomal/Endosomal Transmembrane Protein, Battenin
*CMPK1*	ENSG00000162368	deletion	Cytidine/Uridine Monophosphate Kinase 1
*CROCC*	ENSG00000058453	inversion	Ciliary Rootlet Coiled-Coil, Rootletin
*CYP2C18*	ENSG00000108242	deletion	Cytochrome P450 Family 2 Subfamily C Member 18
*DEFB128*	ENSG00000185982	deletion	Defensin Beta 128
*EFCAB13 **	ENSG00000178852	deletion	EF-Hand Calcium Binding Domain 13
*EIF3C **	ENSG00000184110	deletion	Eukaryotic Translation Initiation Factor 3 Subunit C
*IL18R1 **	ENSG00000115604	inversion	Interleukin 18 Receptor 1
*IL1RL1*	ENSG00000115602	inversion	Interleukin 1 Receptor Like 1
***IL27***	**ENSG00000197272**	**deletion**	**Interleukin 27**
*IL36B*	ENSG00000136696	deletion	Interleukin 36B
*IL37*	ENSG00000125571	deletion	Interleukin 37
*KRTAP19-6*	ENSG00000186925	deletion	Keratin Associated Protein 19-6
*KRTAP19-7*	ENSG00000244362	deletion	Keratin Associated Protein 19-7
*LCN10*	ENSG00000187922	deletion	Lipocalin 10
*LCN6*	ENSG00000267206	deletion	Lipocalin 6
*LGALS14*	ENSG00000006659	deletion	Galectin 14
*MERTK*	ENSG00000153208	deletion	MER Proto-Oncogene, Tyrosine Kinase
*NPIPB8 **	ENSG00000255524	deletion	Nuclear Pore Complex Interacting Protein Family Member B8
*NPIPB9 **	ENSG00000196993	deletion	Nuclear Pore Complex Interacting Protein Family Member B9
*NUPR1 **	ENSG00000176046	deletion	Nuclear Protein 1, Transcriptional Regulator
*OBP2A*	ENSG00000122136	deletion	Odorant Binding Protein 2A
***OR10H1***	**ENSG00000186723**	**deletion**	**Olfactory Receptor Family 10 Subfamily H Member 1**
***OR10H5***	**ENSG00000172519**	**deletion**	**Olfactory Receptor Family 10 Subfamily H Member 5**
*OR2T33*	ENSG00000177212	deletion	Olfactory Receptor Family 2 Subfamily T Member 33
*OR6C2*	ENSG00000179695	deletion	Olfactory Receptor Family 6 Subfamily C Member 2
*OR6C3*	ENSG00000205329	deletion	Olfactory Receptor Family 6 Subfamily C Member 3
*OR6C65*	ENSG00000205328	deletion	Olfactory Receptor Family 6 Subfamily C Member 65
*OR6C70*	ENSG00000184954	deletion	Olfactory Receptor Family 6 Subfamily C Member 70
*OR6C75*	ENSG00000187857	deletion	Olfactory Receptor Family 6 Subfamily C Member 75
*OR6C76*	ENSG00000185821	deletion	Olfactory Receptor Family 6 Subfamily C Member 76
*POU6F2*	ENSG00000106536	deletion	POU Class 6 Homeobox 2
*RABEP2 **	ENSG00000177548	deletion	Rabaptin, RAB GTPase Binding Effector Protein 2
*RACK1*	ENSG00000204628	inversion	Receptor For Activated C Kinase 1
*SGF29 **	ENSG00000176476	deletion	SAGA Complex Associated Factor 29
*SH2B1*	ENSG00000178188	deletion	SH2B Adaptor Protein 1
*SLC35G4*	ENSG00000236396	deletion	Solute Carrier Family 35 Member G4
*SLCO1B3 **	ENSG00000111700	inversion	Solute Carrier Organic Anion Transporter Family Member 1B3
*SULT1A1 **	ENSG00000196502	deletion	Sulfotransferase Family 1A Member 1
*SULT1A2*	ENSG00000197165	deletion	Sulfotransferase Family 1A Member 2
***TUFM***	**ENSG00000178952**	**deletion**	**Tumor Protein P53**
*YAE1D1*	ENSG00000241127	deletion	YAE1 Maturation Factor Of ABCE1
*AC011604.2*	ENSG00000257046	inversion	Uncharacterized
*AL355987.1*	ENSG00000204003	deletion	Uncharacterized

* Human and chimpanzee orthologs were tested and shown to be significant DE genes in either LCLs and/or iPSCs; Genes in bold were found to have strong signatures of positive or balancing selection using the HKA test [45].

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
