# Peer review of "Identification of Structural Variation in Chimpanzees Using Optical Mapping and Nanopore Sequencing"

_genes, 2020, doi:10.3390/genes11030276_

Round 1

Reviewer 1 Report

Soto and colleagues aim to more comprehensively identify deletions and inversions in the chimpanzee genome, and study how this genomic variation alters gene function and regulation.  Interestingly, the authors identify a number of events that appear likely to contribute to chimpanzee-specific differences and present some lovely genome biology.  The method to catalog these regions was well presented, and straightforward use of long-range data types (ie. Optical mapping and nanopore reads).  Overall, I felt the manuscript was well written and the figures were clear for readers.  I have listed only minor comments below to provide some suggestions for the authors on how they may improve their manuscript:

  1. It is true that the human genome is our best reference, and it is reasonable to align chimpanzee sequence data given the expected similarity. With that said, I also recognize that the with long-read sequencing we have seen a great improvement in the chimpanzee reference assembly.  I wonder if the authors would be able to capture another set of variants, missed due to the lack of alignment in hg38, using a PTR reference.
  2. The authors excluded both segmental duplications and insertions from their analysis due to difficulties in discovery and validation.  I think that it would help readers if the authors placed a brief statement in the introduction or in the results describing this -as otherwise, the analysis on first pass seems unbalanced.
  3. It was unclear to me if the authors could tell if the SVs detected were enriched for transposable elements, or specific TE families. TE-based mechanisms have been previously discussed. The issue here is that by aligning to a single human genome the authors may miss a subset of these events (ie PTERV1 described by Yohn et al 2005)
  4. In Figure 1 it looks like the deletions are mapping the human centromere reference models (chr6 and chr20, for example). Do the authors claim to find SVs in these cen/heterochromatic regions too? Also it may be useful to describe why the authors are reporting a deletion on chromosome Y if both cell lines are females? Is this a region similar with the X chromosome and has slightly better mapping to the Y?  

Reviewer 2 Report

I really liked this manuscript by Soto et al. The paper reads very well and the analysis, from what I can gather, has been meticulously conducted. The results introduced many potential targets for future evolutionary studies and, of particular interest, provide a robust set of inversion calls in chimpanzee populations. I thank the authors to compile the paper in such an easily accessible and well-written form. I have only minor comments and some suggestions that should not hinder the publication of this work but may be useful for future work.

Minor comments:

  1. I could not find the information on whether the nanopore reads will be available publically. Please make this clear and if the data will be publically available, provide a link (it is possible that I missed this if so apologies).
  2. I wonder if the dot plots (e.g., Figure 2C) can be replaced by something simpler (the same information can be shown with bar plots for Figure 2C or volcano plots (Figure 3), for example.
  3. The supplementary tables would be very useful for future research. With this in mind, it would be wonderful to provide some explanations for the information provided. For most tables/columns, the information is evident, but there are some columns that may need some explanation.
  4. I understand that the authors did not call duplications - as it still remains a difficult task even with long-reads. This is fine, but I think it should be explicitly stated in the paper.

Some additional suggestions (no need to be addressed, but provided for authors discretion)

  1. We had some interesting observations with regards to copy number differences among different species overlapping conserved regions of the genome supporting the paper's hypothesis (i.e., SV's provide an important role in gene regulation) - I wonder if there is any overlap between these two studies - https://www.pnas.org/content/109/31/12656
  2. I wonder if other comparisons of transcriptomes of chimps and humans from different tissues can provide some further insights? 
  3. I wonder the "mutation rate" - i.e., the occurrence of deletions and inversions overlap with hotspots of CNV formation across different primate species.
  4. It would have been great to have another layer of validation to gather a false-positive false-negative rate - However, I understand that this may be beyond the scope and the authors do provide data from two different platforms (long-read + short-read data). Thus, the data are conservative. Still, it would be interesting to know how well Oxford Nanopore performed for primate genomes for SV detection.

I am Omer Gokcumen by the way - If you have any questions. please let me know ([email protected])

Reviewer 3 Report

Excellent. No suggestions.
This is a well-written paper that needs no editing or changes. Therefore, there is no reason for additional comments.

Author Response

We thank the Reviewer for the kind comments on our study.